# Changes in Physical Function, Cognitive Function, Mental Health, and Sleep Quality After Cardiac Surgeries and Procedures

**DOI:** 10.3390/nursrep15060209

**Published:** 2025-06-11

**Authors:** Yoshimi Kawahara, Nobuto Nakanishi, Keiko Nomura, Satoshi Doi, Jun Oto

**Affiliations:** 1Department of Nursing, Tokushima University Hospital, 2-50-1 Kuramoto, Tokushima 770-8503, Japan; y.kawahara@kangotoku.jp (Y.K.); k.nomura@kangotoku.jp (K.N.); s.doi@kangotoku.jp (S.D.); 2Division of Disaster and Emergency Medicine, Department of Surgery Related, Kobe University Graduate School of Medicine, 7-5-2 Kusunoki, Chuo-Ward, Kobe 650-0017, Japan; 3Emergency and Critical Care Medicine, Tokushima University Hospital, 2-50-1 Kuramoto, Tokushima 770-8503, Japan; joto@tokushima-u.ac.jp

**Keywords:** nurse, cardiac surgery, physical function, cognitive function, mental health problem, sleep quality

## Abstract

**Background**: Patients who undergo cardiac surgery and procedures often experience functional impairments. However, few studies have compared changes in physical function, cognitive function, mental health, and sleep quality before and after the interventions. **Methods**: Intensive care unit (ICU) nurses visited the ward and conducted the assessments. The Japanese version of the Cardiovascular Health Study (J-CHS) and the Barthel index for physical function, mini-mental state examination (MMSE) for cognitive function, hospital anxiety and depression scale for anxiety (HADS-A) and depression (HADS-D) for mental health, and a 5-point Likert scale for sleep quality were used. **Results**: Of the 210 cases, 156 were included. Cardiac surgeries and procedures included valve replacement or valvuloplasty (43%), coronary artery bypass graft (9%), and transcatheter aortic valve implantation (39%). At a median of 7 (4–9) days after ICU discharge, the J-CHS score worsened from 2 (1–3) to 3 (2–3) (*p* < 0.01), and the Barthel index worsened from 95 (85–100) to 75 (55–85) (*p* < 0.01). The HADS-A score improved from 3 (1–6) to 1 (0–4) (*p* < 0.01), and the HADS-D score improved from 4 (1–7) to 2 (1–6) (*p* < 0.01). The MMSE score remained unchanged at 26 (24–29; *p* = 0.91). Sleep quality worsened from 4 (3–5) to 3 (2–4) (*p* < 0.01). In the multivariate analysis, sleep quality deterioration was associated with open thoracotomy. **Conclusions**: After cardiac surgeries and procedures, physical function and sleep quality worsened, whereas anxiety and depression improved, and cognitive function remained unchanged.

## 1. Introduction

Patients in intensive care units (ICUs) experience physical, cognitive, and mental health impairments that persist even after ICU discharge. This condition is referred to as post-intensive care syndrome (PICS). Many patients with PICS are unable to return to their previous lives and jobs. Numerous studies have reported epidemiological data on the incidence of PICS [1,2]. A systematic review and meta-analysis reported a PICS prevalence of 54% [3]. In the surgical ICU, the prevalence was as high as 78% shortly after ICU discharge [4]. However, these studies have primarily focused on the incidence of PICS and do not account for physical, cognitive, and mental health changes from the pre-ICU admission status.

Some older patients may already have impaired functions before ICU admission [5]. To assess newly acquired physical, cognitive, and mental health impairments, changes must be evaluated before and after ICU admission. However, in general ICU settings, assessing these changes is often infeasible because most ICU admissions are emergencies [6]. Additionally, the patient’s condition is frequently unclear upon ICU admission. These condition changes can be better assessed in preplanned surgical admissions [7]. Notably, most cardiac surgeries are scheduled with planned ICU admissions [8]. Patients are typically admitted to the ICU following cardiac surgery. Therefore, in this study, we planned to compare patients’ conditions before and after cardiac surgery.

Patients who undergo cardiac surgery often experience physical impairments, particularly in more severe cases [9]. To our knowledge, few studies have investigated changes in the components of PICS before and after cardiac surgery and procedures. Thus, the present study focused on short-term PICS to observe the effect of cardiac surgery and procedures. Our hypothesis is that patients will develop PICS after undergoing cardiac surgery and procedures with more invasive interventions, such as open thoracotomy, associated with a higher incidence of PICS. To test this hypothesis, both thoracotomy and transcatheter procedures in ICU admissions were included. In addition to physical, cognitive, and mental health problems, changes in sleep quality were included, as PICS is proposed to have various components [10].

## 2. Materials and Methods

### 2.1. Setting

This single-center retrospective study was conducted at the mixed medical/surgical ICU of Tokushima University Hospital. From August 2017 to February 2020, ICU nurses routinely assessed patients’ physical function, cognitive function, mental health, and sleep quality before and after cardiac surgeries and procedures. This retrospective study was approved by the Clinical Research Ethics Committee of Tokushima University Hospital in September 2022. The requirement for individual patient consent was waived, as all data were collected as part of routine clinical practice and retrospectively analyzed. An opt-out method was used, and information about the study was disclosed to patients. It was conducted and reported in accordance with the Strengthening the Reporting of Observational Studies in Epidemiology guidelines [11].

### 2.2. Study Population

This study included consecutive adult patients (aged ≥ 18 years) who were expected to be admitted to the ICU after undergoing cardiac surgery or procedures, such as valve replacement, coronary artery bypass grafting (CABG), and transcatheter aortic valve implantation (TAVI). Patients who declined participation; cases of insufficient staffing for assessment before or after ICU admission; and patients who could not be assessed because of discharge from the hospital, ICU readmission, or death were excluded.

### 2.3. Outcomes

Outcome assessments were conducted by trained ICU nurses, whose assessment skills were approved by a board-certified critical care nurse (Y.K.). Preoperative assessments were performed between hospital admission and the day of surgery or procedure. Postoperative assessments were conducted approximately 7 days after ICU discharge. For study outcomes, the Japanese version of the Cardiovascular Health Study (J-CHS) was used for the assessment of frailty [12], Barthel index for activities of daily living (ADLs) [13], hospital anxiety and depression scale for anxiety (HADS-A) and depression (HADS-D) [14], mini-mental state examination (MMSE) for cognitive function [15], and a 5-point Likert scale for sleep [16,17].

#### 2.3.1. J-CHS

J-CHS is based on the Fried phenotype model [18]. It consists of five items: (1) weight loss, (2) fatigue, (3) low physical activity, (4) slow walking speed, and (5) muscle weakness. (1) Weight loss was defined as a 2 kg loss within 6 months. (2) Fatigue was assessed based on the patient’s feeling of fatigue within 2 weeks. (3) Low physical activity was defined as non-engagement in regular exercise or sports. (4) Slow walking speed was defined as a speed < 1.0 m/s. Participants were asked to walk 10 m at their usual pace, and ICU nurses used a stopwatch to measure the time taken. Gait speed was calculated as “10 m divided by the number of seconds required.” (5) Muscle weakness was defined as decreased grip strength (male < 26 kg, female < 17 kg). Grip strength was measured in a standing position by handgrip dynamometry (TKK 5401, Takei Scientific Instruments Co., Tokyo, Japan). Patients performed two tests on each side, and the maximum value was used for the analysis.

#### 2.3.2. Barthel Index

The Barthel index includes 10 items: (1) feeding, (2) transferring between wheelchair and bed, (3) grooming, (4) using the toilet, (5) bathing, (6) walking on level ground, (7) climbing stairs, (8) dressing, (9) bowel continence, and (10) bladder continence. Higher scores indicate greater independence, whereas lower scores indicate functional impairments.

#### 2.3.3. HADS

HADS is an instrument used to assess anxiety and depression. This 14-item rating scale consists of seven items for anxiety and depression. Responses to each item are scored from 0 (absence of symptoms) to 3 (high prevalence of symptoms). High scores indicate higher levels of symptoms. The lowest possible score is zero, indicating no symptoms, whereas the highest score is 42 (21 for each anxiety and depression domain), indicating severe symptoms.

#### 2.3.4. MMSE

The MMSE includes 10 items assessing various cognitive functions: orientation to time, orientation to place, memorization, calculation, recall, naming, repetition, three-step command, reading comprehension, writing, and structure. The score ranges from 0 to 30, and higher scores indicate better cognitive performance. We purchased and used the Japanese version of the MMSE, whose validity and reliability were demonstrated in a previous study [19].

#### 2.3.5. 5-Point Likert Scale for Sleep

Sleep quality was assessed using a 5-point Likert scale as previously reported [20]. In this scale, a score of 0 indicates no sleep at all; 1, very poor sleep quality; 2, poor sleep; 3, fair sleep; 4, good sleep; and 5, very good sleep.

### 2.4. Delirium

Delirium was assessed using the confusion assessment method for the ICU (CAM-ICU), which evaluates four components: acute changes or fluctuations in mental status, altered level of consciousness, disorganized thinking, and inattention [21]. Nurses assessed delirium three times per shift, and the presence of delirium was recorded if the patient experienced it at least once during their ICU stay.

### 2.5. ICU Mobility Scale (IMS)

ICU nurses conducted daily mobility assessments using the IMS. This tool assesses mobilization capabilities on a scale from 0 (lying in bed) to 10 (walking independently) [22,23]. The maximum IMS score during the ICU stay was used for analysis.

### 2.6. Rehabilitation and Nutrition

Preoperative rehabilitation was not conducted at our facility. Rehabilitation was initiated within 48 h based on a progressive mobilization protocol described by Morris et al. [14]. It was conducted by a multidisciplinary team including bedside nurses and physical therapists. Enteral nutrition was initiated within 48 h and gradually increased based on the patient’s condition. Parenteral nutrition was used only for patients who could not receive enteral nutrition for prolonged periods. No protocolized goals for energy and protein intakes were made.

### 2.7. Multivariate Analysis

Variables in the multivariate analysis were selected based on discussions among board-certified ICU physicians and nurses, considering previously reported relationships with outcome factors. Variables included age, sex, body mass index (BMI), Acute Physiology and Chronic Health Evaluation (APACHE) II score, open thoracotomy, maximum IMS score during the ICU stay, delirium, and physical restraints. Patient characteristics, including age, sex, BMI, and disease severity, are important for multivariate analysis. The invasiveness of surgery was considered an important factor in cardiac surgeries and procedures. Mobilization, delirium, and physical restraints have been previously reported to influence PICS outcomes [24,25]. Cutoff values for the odds ratio were also determined. For age, 65 years is the general retirement age in Japan and is often considered the threshold for being elderly. The BMI threshold of ≥22 kg/m^2^ is an ideal weight reference for evaluating nutritional status [26]. An APACHE II score of ≥20 is reportedly the best cutoff value to predict outcomes [27]. An IMS score of ≥3 is the benchmark for mobilization, as an IMS score of 3 (sitting at the edge of the bed) represents the transition from being bedridden. This threshold of IMS ≥ 3 has been associated with improved functional outcomes [28].

### 2.8. Sample Size and Statistical Analyses

The sample size was not calculated a priori, as this was a retrospective study based on existing clinical records. Continuous data were expressed as mean ± standard deviation for normally distributed variables or median (interquartile range [IQR]) for non-normally distributed variables, whereas categorical data were expressed as a number (%). Continuous variables were analyzed using the t-test or Mann–Whitney U test, whereas categorical variables were analyzed using the chi-square test. Functional outcomes before and after cardiac surgeries and procedures were compared using the Wilcoxon signed-rank test, as paired data did not follow a normal distribution. Missing data were not imputed. Data analyses were conducted using JMP version 13.1.0 (SAS Institute Inc., Cary, NC, USA). All statistical tests were two-tailed, with a type 1 error rate set at *p* < 0.05.

## 3. Results

During the study period, a total of 210 patients underwent cardiac surgery (Figure 1), and 198 patients were assessed preoperatively. Of these patients, 42 were excluded, leaving 156 in the analysis. Patients’ mean age was 74 ± 16 years, and 64 (41%) were male (Table 1). The median APACHE II score was 14 (12–17). The median ICU stay was 4 (2–5) days. The types of cardiac surgery included valve replacement or valvuloplasty (43%), CABG (9%), a combination of valve replacement and CABG (5%), and TAVI (39%). Patients were assessed a median of 7 (4–9) days after ICU discharge.

Among the 156 patients, data were available for 155 on the J-CHS, 156 on the Barthel Index, 154 on HADS-A, 154 on HADS-D, 155 on the MMSE, and 153 on the sleep quality assessment. Patients’ conditions changed from preoperative to postoperative assessments (Figure 2). The J-CHS indicated a worsened status in 56 (36%) patients, with the median score changing from 2 (1–3) to 3 (2–3) (*p* < 0.01). The Barthel index indicated a worsened status in 60 (39%) patients, with the median score changing from 95 (85–100) to 75 (55–85) (*p* < 0.01). The HADS-A indicated improvement in 93 (60%) patients, remained unchanged in 22 (14%), and worsened state in 39 (25%), with the median HADS-A score changing from 3 (1–6) to 1 (0–4) (*p* < 0.01). The HADS-D indicated improvement in 79 (51%) patients, remained unchanged in 30 (20%), and worsened state in 45 (29%), with the median HADS-D score changing from 4 (1–7) to 2 (1–6) (*p* < 0.01). The MMSE indicated deterioration in 60 (39%) patients, though the median score remained unchanged at 26 (24–29) (*p* = 0.91). Sleep quality deteriorated in 70 (46%) patients, with the median score changing from 4 (3–5) to 3 (2–4) (*p* < 0.01). Detailed changes in each score are shown in Appendix A.

Risk factor analysis was conducted for factors associated with physical and sleep quality deterioration (Table 2). Age ≥ 65 years, sex, BMI ≥ 22 kg/m^2^, APACHE II score ≥ 20, open thoracotomy, maximum IMS ≥ 3, delirium, and physical restraint were not associated with J-CHS and Barthel index deterioration in the univariate and multivariate analyses. Open thoracotomy, maximum IMS ≥ 3, and delirium were associated with sleep quality deterioration in the univariate analysis (thoracotomy, 3.77 [CI, 1.89–7.52]; maximum IMS ≥ 3, 2.95 [CI, 1.10–7.92]; delirium, 2.13 [CI, 1.11–4.09]). However, only open thoracotomy was associated with sleep quality deterioration in the multivariate analysis (thoracotomy, 3.43 [CI, 1.37–8.62]).

## 4. Discussion

This study revealed that physical function, ADLs, and sleep quality worsened after cardiac surgeries and procedures, whereas anxiety and depression improved, and cognitive function remained unchanged. This study compared patient conditions before and after cardiac surgeries and procedures, thus eliminating population bias. Few risk factors were identified for physical and ADL deterioration, whereas open thoracotomy was associated with impaired sleep quality. Patient characteristics, mobilization, physical restraints, and delirium were not associated with these functional changes, indicating that numerous factors influence these changes after cardiac surgeries and procedures.

PICS after intensive care is a frequently discussed issue. PICS encompasses physical, cognitive, and mental domains. Interestingly, these three aspects changed differently after cardiac surgeries and procedures. In this study, physical function, including frailty and ADLs, worsened, whereas mental health improved after cardiac surgeries and procedures. Conversely, cognitive function did not change. These differences are important data for patient management.

In the present study, physical function declined in terms of frailty and ADLs. A previous study reported that physical function declined in 17% of patients who underwent cardiac surgery [29]. Although the present study included less invasive procedures such as TAVI, physical impairments were observed in 54% of the patients for frailty and 36% for ADLs. This finding is noteworthy. In the multivariate analysis, neither thoracotomy nor non-thoracotomy was associated with physical impairments. Although TAVI is a less invasive procedure, the majority of patients still experience physical impairments. The average age was significantly higher in patients who underwent TAVI than in those who underwent thoracotomy (TAVI vs. thoracotomy: 84 ± 2 vs. 67 ± 1, *p* < 0.01). The older age of patients undergoing TAVI may contribute to greater physical impairments, although TAVI is a less invasive procedure.

In this study, anxiety and depression improved after ICU discharge. This result contrasts with the finding of a previous study [30], where 41% of the patients reported anxiety and 22% reported depression 12 weeks after hospital discharge. Our results may differ because many patients have concerns before cardiac surgery or procedure, which dissipate afterward. Therefore, the tendency of anxiety and depression to decrease in preplanned admission cases must be considered. However, the long-term mental health effects following hospital discharge remain unclear. Physical impairments often persist after discharge, and the associated stress and economic burden may contribute to long-term mental health problems.

Cognitive function did not differ before and after cardiac surgeries and procedures. Delirium was observed in 71 (72%) patients at least once during their ICU stay, with a median duration of 1 (0–2) days. A study found that delirium was associated with impaired cognitive function after hospital discharge [31]. However, in the present study, cognitive function did not change despite the occurrence of delirium. This study focused on patients undergoing cardiac surgery and procedures, whereas many other studies included patients with sepsis, which can cause septic encephalopathy [32,33]. Neurological damage may occur in certain inflammatory conditions [34]. However, cognitive function generally may not change in preplanned cardiac surgeries and procedures. This result is contrary to those of previous studies [35,36]. The difference may be in the assessment timing. Cognitive outcome was assessed at approximately 7 days after ICU discharge, which may not fully capture subtle or fluctuating impairments that follow the high prevalence of delirium.

In this study, sleep quality worsened after cardiac surgeries. A study highlighted prolonged sleep problems as an important aspect of PICS. During ICU stays, patients experience low sleep quality [37], and this sleep disturbance may continue even after ICU discharge. Various factors cause prolonged sleep disturbance including psychological stress, delirium, medication withdrawal, environmental changes, and disrupted circadian rhythms [38]. One of the important causes of impaired sleep quality is postoperative pain [39]. The results of the present study suggested that thoracotomy was associated with low sleep quality, likely due to pain or the invasiveness of the surgery. Although sleep deterioration following thoracotomy is likely related to postoperative pain, pain levels were not systematically assessed in non-intubated patients in this study. This limitation should be considered when interpreting the results. Further studies are warranted to explore the causes and potential prevention of sleep problems in patients who underwent thoracotomy.

Our study findings have important implications for nursing care. Preoperative nursing assessments can help identify patients at high risk of developing PICS. As our results demonstrated significant declines in physical function, ADL, and sleep quality following the intervention, patients with pre-existing impairments in these domains may benefit from early and intensive interventions [40,41]. Sharing this information among nursing staff, as well as rehabilitation and nutrition teams, will facilitate multidisciplinary collaboration and enhance care during the ICU stay [42]. Moreover, nursing assessments following ICU discharge can support continuity of care for patients with functional decline. Those who experience deterioration in functional status should receive ongoing support and follow-up care post-ICU [43]. This information is therefore essential for ensuring continuity and comprehensiveness of care. In particular, sleep disturbances can be addressed through nursing interventions such as stress management, therapeutic communication, and referral to specialists [44]. This is especially relevant for patients undergoing thoracotomy, who were found to be at greater risk for postoperative sleep deterioration.

This study has several limitations. First, the sample size was not calculated a priori. However, in multivariable analysis, we ensured that the number of events per variable (EPV) met commonly accepted standards. Specifically, the number of events in our outcome variable and the number of covariates included in the model allowed for an EPV of at least 10, which is generally regarded as sufficient to minimize the risk of overfitting and to ensure stable estimates. Second, as this was a retrospective study based on routinely collected clinical data, further prospective studies are warranted to validate our findings. Third, long-term changes were not assessed. This study focused on short-term PICS after ICU discharge because long-term outcomes are influenced by numerous factors, such as economic situation and PICS follow-up clinics [45]. Fourth, clear risk factors for physical impairments were not identified. Some of our findings differ from those of previous studies. For example, the maximum IMS score was reported as a predictor of long-term patient outcomes [24]. However, in the present study, the maximum IMS score was not associated with physical impairments. The IMS score reflects the mobilization level rather than physical function and is often affected by staff shortages or scheduling constraints, making this finding reasonable. Fifth, pain was only assessed using behavioral pain scale; thus, pain in patients without mechanical ventilation support was not clearly evaluated. Sixth, both surgical and catheter treatments, which are different interventions, were included as they needed to be analyzed together in this study to compare their effects. Furthermore, the surgical approaches used in surgical treatment were not differentiated owing to the wide variety of interventions. Seventh, assessment was conducted approximately 7 days after ICU discharge. Therefore, patients’ condition at that time of assessment may have been influenced by the length of ICU stay.

## 5. Conclusions

In this study, changes in physical function, cognitive function, mental health, and sleep quality were investigated before and after cardiac surgeries and procedures. The results reveal that physical function and sleep quality worsened, whereas anxiety and depression improved, and cognitive function remained unchanged. Sleep quality deterioration was associated with thoracotomy; however, no specific risk factors were identified for physical impairments.

## Figures and Tables

**Figure 1 nursrep-15-00209-f001:**
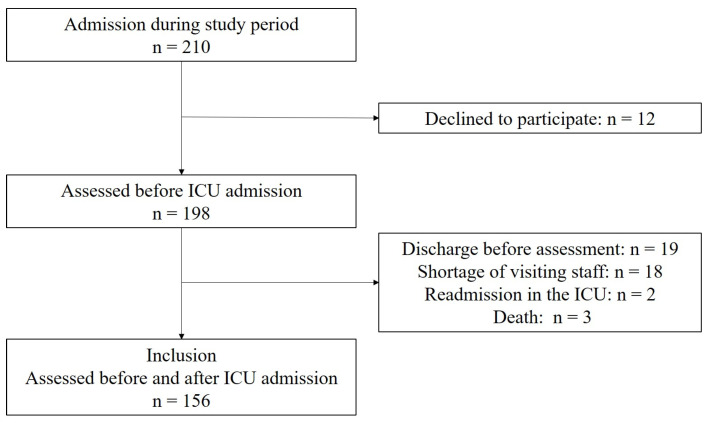
Flow of patients in the study.

**Figure 2 nursrep-15-00209-f002:**
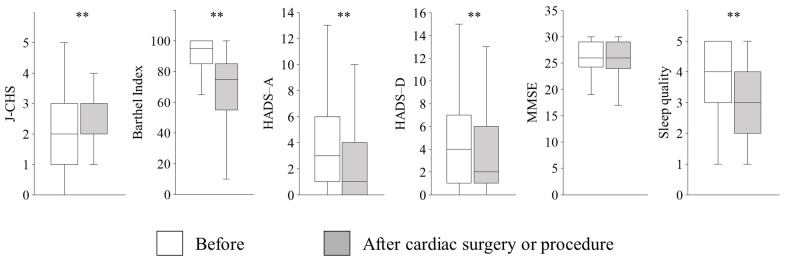
Changes in physical function, cognitive function, mental health, and sleep quality before and after cardiac surgeries and procedures. The J-CHS indicated a worsened status, with the score changing from 2 (1–3) to 3 (2–3) (*p* < 0.01). The Barthel index indicated a worsened status, with the score changing from 95 (85–100) to 75 (55–85) (*p* < 0.01). The HADS-A presented improvement, with the score changing from 3 (1–6) to 1 (0–4) (*p* < 0.01). The HADS-D suggested improvement, with the score changing from 4 (1–7) to 2 (1–6) (*p* < 0.01). The MMSE score remained unchanged at 26 (24–29) (*p* = 0.18). Sleep quality deteriorated, with the score changing from 4 (3–5) to 3 (2–4) (*p* < 0.01). Data were expressed as median (IQR) and analyzed using the Wilcoxon signed-rank test. HADS-A, hospital anxiety and depression scale for anxiety; HADS-D, hospital anxiety and depression for depression; J-CHS, Japanese version of the Cardiovascular Health Study; MMSE, mini-mental state examination. ** Significant at *p* < 0.01.

**Table 1 nursrep-15-00209-t001:** Patient characteristics.

	All Patients
Variables	n = 156
Age, mean ± SD, y	74 ± 16
Male/female	64/92
Body mass index, mean ± SD, kg/m^2^	23.1 ± 3.9
APACHE II score	14 (12–17)
SOFA	5 (3–7)
Treatment type, n (%)	
Valve replacement or valvuloplasty	67 (43)
CABG	14 (9)
Combination of valve and CABG	6 (4)
TAVI	60 (39)
Others	9 (6)
Thoracotomy, n (%)	89 (57)
Mechanical ventilation, n (%)	99 (64)
Length of ICU stay	4 (2–5)
Comorbidities, n (%)	
High blood pressure	103 (66)
Diabetes mellitus	44 (28)
Stroke	28 (18)
Liver disease	11 (7)
Kidney disease	49 (31)
Lung disease	16 (1)

Data are presented as medians (IQR) or number (percentage) unless otherwise indicated. APACHE, Acute Physiology and Chronic Health Evaluation; CABG, coronary artery bypass grafting; ICU, intensive care unit; IQR, interquartile range; SD, standard deviation; SOFA, Sequential Organ Failure Assessment; TAVI, transcatheter aortic valve implantation.

**Table 2 nursrep-15-00209-t002:** Univariate and multivariate analyses for J-CHS, Barthel index, and sleep quality.

	J-CHS Deterioration		Barthel Index Deterioration		Sleep Quality Deterioration	
Variables	Univariate	Multivariate	Univariate	Multivariate	Univariate	Multivariate
Age ≥ 65 years	1.85 (0.75–4.57)	1.91 (0.68–5.40)	0.78 (0.27–2.25)	0.95 (0.28–3.21)	0.89 (0.39–2.05)	1.67 (0.60–4.65)
Male/female	0.93 (0.42–2.06)	1.09 (0.46–2.58)	1.61 (0.72–3.58)	1.62 (0.70–3.76)	1.49 (0.78–2.85)	1.20 (0.58–2.51)
Body mass index ≥ 22 kg/m^2^	1.15 (0.51–2.60)	1.05 (0.44–2.55)	1.32 (0.62–2.85)	1.13 (0.51–2.52)	1.90 (0.98–3.67)	1.57 (0.77–3.22)
APACHE II score ≥ 20	0.73 (0.26–2.08)	0.60 (0.20–1.82)	6.09 (0.79–47.23)	5.99 (0.75–48.15)	0.97 (0.38–2.48)	0.59 (0.19–1.79)
Thoracotomy	0.80 (0.24–2.72)	0.84 (0.24–2.98)	1.95 (0.91–4.21)	1.68 (0.63–4.50)	**3.77 (1.89–7.52)**	**3.43 (1.37–8.62)**
Maximum IMS ≥ 3	0.76 (0.12–4.74)	0.77 (0.11–5.40)	1.90 (0.74–4.88)	1.47 (0.52–4.20)	**2.95 (1.10–7.92)**	1.75 (0.58–5.23)
Delirium	1.49 (0.62–3.58)	1.28 (0.48–3.47)	1.26 (0.59–2.70)	0.86 (0.35–2.12)	**2.13 (1.11–4.09)**	1.28 (0.56–2.95)
Physical restraints	1.17 (0.25–5.52)	1.28 (0.24–6.94)	0.68 (0.13–3.69)	0.58 (0.09–3.68)	0.88 (0.19–4.09)	0.48 (0.09–2.70)

Data are presented as medians (IQR). Bold numbers indicate the significant differences. J-CHS, Japanese version of the Cardiovascular Health Study criteria; APACHE II, Acute Physiology and Chronic Health Evaluation.

## Data Availability

Data cannot be shared publicly due to institutional ethics and confidentiality regulations.

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
