# Peer review of "Changes in Physical Function, Cognitive Function, Mental Health, and Sleep Quality After Cardiac Surgeries and Procedures"

_nursrep, 2025, doi:10.3390/nursrep15060209_

Round 1
Reviewer 1 Report
Comments and Suggestions for Authors
This manuscript addresses a relevant topic in critical care nursing by evaluating changes in physical, cognitive, and psychological outcomes before and after cardiac surgeries. The findings are clinically valuable and methodologically sound. However, the manuscript requires the following minor but mandatory revisions:
- Reporting Standards:
Please include a statement confirming adherence to STROBE guidelines for observational studies. This is essential for transparency and completeness in reporting. - Justification of Variables in Multivariate Analysis:
Some dichotomization thresholds (e.g., BMI ≥22, IMS ≥3) need clearer rationale, supported by literature or clinical standards. - Delirium vs Cognitive Outcomes:
The discussion should further explore the unexpected lack of association between delirium and cognitive decline, especially given the high delirium rate (72%). - Pain Assessment:
Sleep deterioration is attributed to thoracotomy, likely due to postoperative pain. However, pain was not systematically assessed in non-intubated patients. This limitation should be explicitly acknowledged.
Several sections would benefit from revision for grammar, clarity, and academic tone. A professional English language edit is strongly recommended before publication
Author Response
Comments 1:
This manuscript addresses a relevant topic in critical care nursing by evaluating changes in physical, cognitive, and psychological outcomes before and after cardiac surgeries. The findings are clinically valuable and methodologically sound. However, the manuscript requires the following minor but mandatory revisions:
Response 1:
Thank you for the reviewer’s positive comment.
Comments 2:
Reporting Standards:
Please include a statement confirming adherence to STROBE guidelines for observational studies. This is essential for transparency and completeness in reporting.
Response 2:
It is important point to be added. We checked that our manuscript adhered to STROBE guidelines. We added as following. L72 “It was conducted and reported in accordance with the Strengthening the Reporting of Observational Studies in Epidemiology guidelines [11]”
Comments 3:
Justification of Variables in Multivariate Analysis:
Some dichotomization thresholds (e.g., BMI ≥22, IMS ≥3) need clearer rationale, supported by literature or clinical standards.
Response 3:
Thank you for your valuable comment. We agree that the rationale for dichotomization thresholds should be clearly justified. We have revised the manuscript to include references and explanations for the selected cut-off values as follows: BMI ≥22: This threshold is aligned with the criteria of malnutrition in clinical nutrition guidelines [PMID: 30181091]. ICU Mobility Scale (IMS) ≥3: An IMS score of 3 or higher has been used in previous literature to indicate the achievement of out-of-bed mobilization, which is associated with improved functional outcomes and reduced ICU-acquired weakness [PMID: 35468868]. The manuscript was revised as following: L166 “The BMI threshold of ≥22 kg/m² is an ideal weight reference for evaluating nutritional status [25].” L170 “This threshold of IMS ≥3 has been associated with improved functional outcomes [27].”
Comments 4:
Delirium vs Cognitive Outcomes: The discussion should further explore the unexpected lack of association between delirium and cognitive decline, especially given the high delirium rate (72%).
Response 4:
Thank you for this insightful comment. We agree that the lack of a significant association between delirium and cognitive decline in our findings is unexpected, particularly given the high incidence of delirium (72%) in our cohort. We have revised the Discussion section to address this point more thoroughly as following: L307 “This result is contrary to those of previous studies [34,35]. The difference may be in the assessment timing. Cognitive outcome was assessed at approximately 7 days after ICU discharge, which may not fully capture subtle or fluctuating impairments that follow the high prevalence of delirium.”
Comments 5:
Pain Assessment: Sleep deterioration is attributed to thoracotomy, likely due to postoperative pain. However, pain was not systematically assessed in non-intubated patients. This limitation should be explicitly acknowledged.
Response 5:
Thank you for pointing this out. We agree with the reviewer that the lack of systematic pain assessment in non-intubated patients is a limitation of our study. We have now explicitly acknowledged this in the Discussion section and have revised the text as follows: L319 “Although sleep deterioration following thoracotomy is likely related to postoperative pain, pain levels were not systematically assessed in non-intubated patients in this study. This limitation should be considered when interpreting the results. Further studies are warranted to explore the causes and potential prevention of sleep problems in patients who underwent thoracotomy.”
Reviewer 2 Report
Comments and Suggestions for Authors
Dear authors,
The topic addressed in this manuscript is of considerable relevance to the field of nursing and to healthcare more broadly, particularly given the growing interest in improving patient outcomes through evidence-based practices. However, the manuscript presents several limitations that should be addressed in order to strengthen its scientific rigor and overall clarity.
One of the main weaknesses lies in the Introduction and Discussion sections, which lack a solid theoretical foundation. The literature review is insufficient and does not adequately frame the research problem within the broader context of current knowledge. Identification of previous studies that either support or contradict the results presented, particularly in light of the various surgical approaches and procedures examined in this study.
Moreover, some key aspects of the topic are not sufficiently explored or developed. Certain concepts and findings require deeper analysis and should be discussed in greater depth. The contextualization of the results is limited, and the discussion does not always clearly link the findings back to the research aim. Strengthening these elements would greatly enhance the interpretability and relevance of the study.
The Methods section also requires further clarification. Important methodological details are either missing or not described with enough precision to allow full understanding of the procedures used.
Finally, the presentation of the results could be improved. As currently structured, the results lack come clarity and coherence. Organizing the data more systematically and providing a clearer table would help in this regard.
In summary, while the article addresses a pertinent and timely issue within nursing and healthcare, revisions are necessary to improve its theoretical grounding, methodological transparency, and the depth of its discussion. These improvements would significantly enhance the manuscript’s overall quality and impact.

Author Response
Comment 1:
The topic addressed in this manuscript is of considerable relevance to the field of nursing and to healthcare more broadly, particularly given the growing interest in improving patient outcomes through evidence-based practices. However, the manuscript presents several limitations that should be addressed in order to strengthen its scientific rigor and overall clarity. One of the main weaknesses lies in the Introduction and Discussion sections, which lack a solid theoretical foundation. The literature review is insufficient and does not adequately frame the research problem within the broader context of current knowledge. There is a need for a more comprehensive and critical engagement with existing literature to justify the study's objectives and to situate the findings within ongoing academic and clinical knowledge. This includes the identification of previous studies that either support or contradict the results presented, particularly in light of the various surgical approaches and procedures examined in this study. Moreover, some key aspects of the topic are not sufficiently explored or developed. Certain concepts and findings require deeper analysis and should be discussed in greater depth. The contextualization of the results is limited, and the discussion does not always clearly link the findings back to the research aim. Strengthening these elements would greatly enhance the interpretability and relevance of the study. The Methods section also requires further clarification. Important methodological details are either missing or not described with enough precision to allow full understanding of the procedures used. Finally, the presentation of the results could be improved. As currently structured, the results lack come clarity and coherence. Organizing the data more systematically and providing clearer tables would help in this regard. In summary, while the article addresses a pertinent and timely issue within nursing and healthcare, revisions are necessary to improve its theoretical grounding, methodological transparency, and the depth of its discussion. These improvements would significantly enhance the manuscript’s overall quality and impact.
Response 1:
Thank you very much for your thoughtful and comprehensive feedback. We appreciate your recognition of the relevance of our study to the field of nursing and healthcare, as well as your detailed suggestions for improvement. We have carefully revised the manuscript to address each comment as outlined below. We have significantly revised the Introduction and Discussion sections to incorporate a stronger theoretical framework. We expanded the literature review to more comprehensively present current knowledge in the field, including recent studies relevant to PICS and surgical ICU outcomes. This provides a clearer justification for our study’s aims and helps situate our findings within a broader academic and clinical context. Key concepts and findings that were previously underdeveloped have now been elaborated upon. We expanded the Discussion to better interpret our results in light of existing evidence and clearly linked our findings back to the research objectives. The Methods section has been revised for clarity and completeness. We included more precise descriptions of our study. We have reorganized the results section for better coherence and readability. The tables have been revised to present data more clearly.
Comment 2:
Abstract L23 – You refer 7 days after ICU discharge. Mean? Please state clearly which statistical measure you are referring to.
Response 2:
It was median. We added as following. L23 “At a median of 7 (4-9) days after ICU discharge”
Comment 3:
Introduction L37 – English “is referred to as Post-Intensive Care Syndrome (PICS)”
Response 3:
We revised the point.
Comment 4:
L40 – It may be import for you to consider that only two studies are presented referring to PICS. Literature offers more descriptive information on these specifics conditions after surgery. Perhaps a stronger theoretical frame would help justify the need for you study
Response 4:
We acknowledge that only two studies directly referring to PICS (Post-Intensive Care Syndrome) were included in our analysis. There is a scarcity of literature in our background. We identified additional literature describing postoperative cognitive, physical, and psychological impairments. We revised as following. L40 “A systematic review and meta-analysis reported a PICS prevalence of 54% [3]. In the surgical ICU, the prevalence was as high as 78% shortly after ICU discharge [4].”
Comment 5:
L42 – Doesn’t this reflection belong at the introduction end after all previous existing knowledge on the topic is presented?
Response 5:
We agree. We moved the part to the next paragraph.
Comment 6:
L44-51 – Most major cardiac elective surgeries require an ICU stay of at least 24h. Introduction is supposedly a review of existing literature regarding a topic, there´s lacking literature to support your claims.
Response 6:
We added the literature to support our claims in the introduction.
Comment 7:
L53 – “In more severe cases”, which cases are more severe? The surgery itself? The previous condition of the patient? Lacks contexts.
Response 7:
We meant more invasive intervention. We revised as following L58 “Our hypothesis is that patients will develop PICS after undergoing cardiac surgery and procedures with more invasive interventions, such as open thoracotomy, associated with a higher incidence of PICS”.
Comment 8:
L57 – Please provide context to not include sternotomy in your study. Consider presenting the rationale related to prevalence of all known approaches to heart surgery, and reflect on this in the discussion/study limitations.
Response 8:
Thank you for your valuable comment. We agree that it is important to clarify the approaches to heart surgery. Our study included various surgery and operation including open thoracotomy and TAVI for the purpose to reveal the change of function. Due to the various background with surgery and procedure, we did not analyze the detail of approach such as minimally invasive cardiac surgery. However, as reviewer mentions, it should be included in the limitation. We added as following: L337 “Furthermore, the surgical approaches used in surgical treatment were not differentiated owing to the wide variety of interventions”.
Comment 9:
Study population L74 – Patients who had epicardial pacemaker or other implemented devices during surgery were included or excluded?
Response 9:
We did not exclude these patients in our study.
Comment 10:
L76 – Patients cognitively impaired preoperatively were included? If yes, explain rationale.
Response 10: We did not exclude patients cognitively impaired preoperatively because we assessed the change of cognitive function, not the existence of cognitive impairments or not.
Comment 11:
Outcomes: L84 – After one week after ICU discharge may influence the results of your study because the length of days in these specific situations have an impact on how the patient feels and may compromise the report. One week meaning 5 or 7 days? Before hospital discharge should be clearly stated.
Response 11:
One week meant 7 days. We conducted the assessment approximately 7 days of ICU discharge. It was not exactly 7 days due to the shortage of staff, weekend, and other barrier to measure exactly in the same day. As mentioned by the reviewer, the length of days in these specific situations may have an impact on how the patient feels and may compromised the report. We added these points as following. Method: “Postoperative assessments were conducted around 7 days after ICU discharge”. Limitation: “Fifth, assessment was conducted approximately 7 days after ICU discharge. Therefore, patients’ condition at that time of assessment may have been influenced by the length of ICU stay”.
Comment 12:
L90-140 - Authors of the instruments (if translated to Japanese as well) should be cited throughout the text. Some of the instruments even if property of a company still has authors.
Response 12:
We added the citation of the instruments throughout the manuscript.
Comment 13:
Results L201-202 – the numbers in parenthesis are percentages? Please proceed to correct.
Response 13:
We revised the point.
Comment 14:
L203 – Length of ICU stays are often presented in hours. I believe you present in days. Different data is presented in this table, rethink the presentation of this information on the table.
Response 14: It is important point to present the data of length of ICU stay in hours. However, we do not have the data regarding hours. Therefore, we presented the data of days.
Comment 15: Please justify why you chose to use median to present the results instead of means.
Response 15: We used median because the data were not normal distribution. Regarding age and BMI, we used mean because the data were normal distribution.
Comment 16:
Discussion L270 – This reflexion can be deepened. Are there any relevant sociodemographic characteristics in the TAVI population compared to the ones submitted to thoracotomy such as age? Can you explore in detail the differences between patients proposed to TAVI and open-heart surgery?
Response 16:
We analyzed the difference in patient sociodemographic characteristics in the TAVI and thoracotomy. We found there was a difference in age. Patients who underwent TAVI had much older age than thoracotomy. Therefore, TAVI may have been associated with impaired physical impairments, although it is less invasive procedure. We added as following. L285 “The average age was significantly higher in patients who underwent TAVI than in those who underwent thoracotomy (TAVI vs. thoracotomy: 84 ± 2 vs. 67 ± 1, p < 0.01). The older age of patients undergoing TAVI may contribute to greater physical impair-ments, although TAVI is a less invasive procedure.”
Comment 17:
L285 – Referring studies that are not cited.
Response 17:
We added the citation as following. L303 “This study focused on patients undergoing cardiac surgery and procedures, whereas many other studies included patients with sepsis, which can cause septic encephalopathy [31,32]. Neurological damage may occur in certain inflammatory conditions [33].”
Comment 18:
L291 – There is only one reason for sleep impairment? Only one study to support such claim.
Response 18:
As mentioned, sleep impairment is caused by numerous causes. We added the discussion as following. L315 “Various factors cause prolonged sleep disturbance including psychological stress, delirium, medication withdrawal, environmental changes, and disrupted circadian rhythms [37].”
Comment 19:
L300 – You present a rationale but not the author of the information.
Response 19:
We added the information as following. L329 “For example, the maximum IMS score was reported as a predictor of long-term patient outcomes [23].”
Comment 20:
L306 – English “Which are different interventions”
Response 20:
We revised the point.
Round 2
Reviewer 2 Report
Comments and Suggestions for Authors
Firstly, congratulation on your now improved paper. I present some remarks below:
Introduction is now fairly grounded in theory.
Methodology: Ethical aspects are not mentioned in the manuscript. The authors should clarify whether informed consent was obtained, whether ethical approval from the hospital’s ethics committee was granted, even if its states at the end that “Informed Consent Statement: Patient consent was waived due to the retrospective nature of this study”, ethical considerations should always be referred as they are a main key in every research. Also you should mention if permission was obtained to use the instruments, especially the Mini-Mental State Examination (copyrighted and requires purchase).
Results and discussion are fairly improved. Although some questions remain unclear, as why median was used to report the results.
Author Response
Comment 1: Firstly, congratulation on your now improved paper. I present some remarks below: Introduction is now fairly grounded in theory.
Response 1: Thank you for the positive comment. We are encouraged by your comment.
Comment 2: Methodology: Ethical aspects are not mentioned in the manuscript. The authors should clarify whether informed consent was obtained, whether ethical approval from the hospital’s ethics committee was granted, even if its states at the end that “Informed Consent Statement: Patient consent was waived due to the retrospective nature of this study”, ethical considerations should always be referred as they are a main key in every research.
Response 2: We agree. We should state that in the method section. We added as following. “The requirement for each patient consent was waived due to the retrospective design of the study.”
Comment 3: Also you should mention if permission was obtained to use the instruments, especially the Mini-Mental State Examination (copyrighted and requires purchase).
Response 3: Yes. We purchased the Japanese version of the MMSE from Okada Synthesis Psychology Center Ltd. as a distributor of Nihon Bunka Kagakusha Co. The Japanese version of MMSE-J is owned by Nihon Bunka Kagaku-sha under license from PAR, Inc. In the method section, we added as following. “We purchased and used the Japanese version of the MMSE, whose validity and reliability were demonstrated in a previous study [19].”
Comment 4: Results and discussion are fairly improved. Although some questions remain unclear, as why median was used to report the results.
Response 4: Our explanation was insufficient. We used the median because the data were non-normally distributed variables. We revised the method section as following “Continuous data were expressed as mean ± standard deviation for normally distributed variables or median (interquartile range [IQR]) for non-normally distributed variables.”